# IOT: INSTANCE-WISE LAYER REORDERING FOR TRANSFORMER STRUCTURES

**Jinhua Zhu**[1,*], **Lijun Wu**[2,*], **Yingce Xia**[2], **Shufang Xie**[2],
**Tao Qin**[2], **Wengang Zhou**[1], **Houqiang Li**[1], **Tie-Yan Liu**[2]
[1]University of Science and Technology of China;
[2]Microsoft Research;
[1]teslazhu@mail.ustc.edu.cn, {zhwg,lihq}@ustc.edu.cn,
[2]{Lijun.Wu,Yingce.Xia,shufxi,taoqin,tyliu}@microsoft.com

## ABSTRACT

With sequentially stacked self-attention, (optional) encoder-decoder attention, and feed-forward layers, Transformer achieves big success in natural language processing (NLP), and many variants have been proposed. Currently, almost all these models assume that the *layer order* is fixed and kept the same across data samples. We observe that different data samples actually favor different orders of the layers. Based on this observation, in this work, we break the assumption of the fixed layer order in Transformer and introduce instance-wise layer reordering into model structure. Our Instance-wise Ordered Transformer (IOT) can model variant functions by reordered layers, which enables each sample to select the better one to improve the model performance under the constraint of almost same number of parameters. To achieve this, we introduce a light predictor with negligible parameter and inference cost to decide the most capable and favorable layer order for any input sequence. Experiments on 3 tasks (neural machine translation, abstractive summarization, and code generation) and 9 datasets demonstrate consistent improvements of our method. We further show that our method can also be applied to other architectures beyond Transformer. Our code is released at Github[1].

## 1 INTRODUCTION

Transformer (Vaswani et al., 2017) has been the dominant architecture in deep learning models (Hassan et al., 2018; Ng et al., 2019; Carion et al., 2020; Radford et al., 2019; Dai et al., 2019; Lee et al., 2019; Devlin et al., 2018; Yang et al., 2019; Cai & Lam, 2019). A Transformer model is stacked by several identical blocks, and each block consists of sequentially ordered layers: the self-attention (*SA*), encoder-decoder attention (*ED*) (decoder only) and feed-forward (*FF*) layer. Recently, various modifications have been proposed, where the focus is on replacing or inserting some components (e.g., attention layer/layer norm/position encoding) in standard Transformer (Wu et al., 2019; Lu et al., 2019; Shaw et al., 2018; So et al., 2019; Ahmed et al., 2017).

Despite these Transformer alternatives have achieved improved performances, one critical element is almost neglected in current models, which is how to arrange the components within a Transformer network, i.e., the *layer order* also matters. As pointed by He et al. (2016b), different orders of ReLU, batch normalization and residual connection significantly affect the performance of ResNet (He et al., 2016a). Therefore, we ask: What if we reorder the sequential layers in Transformer (e.g., *SA→FF* or *FF→SA* of encoder, *SA→FF→ED* or *FF→ED→SA* of decoder)? What is the best order for these different layers?

| Order | De→En | Ratio |
|---|---|---|
| *1:SA→ED→FF* | 34.64 | 17.9% |
| *2:FF→SA→ED* | 34.60 | 17.9% |
| *3:ED→FF→SA* | 34.75 | 16.5% |
| *4:ED→SA→FF* | 34.67 | 17.5% |
| *5:SA→FF→ED* | 34.76 | 14.1% |
| *6:FF→ED→SA* | 34.78 | 16.1% |
| Variance | 0.0045 | - |

Table 1: Results for different decoder orders on IWSLT14 De→En translation.

---

*Equal contribution and corresponding authors.
[1]https://github.com/instance-wise-ordered-transformer/IOT

| Reference | and just like that , the iceberg shows you a different side of its personality . | BLEU↑ | TER↓ |
|---|---|---|---|
| Order 1 Trans | and just so , the iceberg shows a different side of its personality . | 77.11 | 18.75 |
| Order 2 Trans | and just like that , the iceberg shows you a different side of its personality . | 100.00 | 0.00 |
| Order 3 Trans | and just so , the iceberg gives you another side of his personality . | 0.00 | 37.50 |
| Order 4 Trans | and just like this , the iceberg gives you another side of its personality . | 38.71 | 25.00 |
| Order 5 Trans | ans so simply , the iceberg shows another side of his personality . | 30.33 | 50.00 |
| Order 6 Trans | and just like this , the iceberg shows you another side of his personality . | 36.61 | 25.00 |

Table 2: Translations (*Trans*) from all ordered decoders of Transformer for one example sentence.

We first conduct preliminary experiments. We vary the three layers in decoder with all six variants (each with a unique order of the three layers) and train these models. Results on IWSLT14 German→English translation are reported in Table 1. As we can see, their performances are similar and no one is outstanding. The corpus BLEU variance is only 0.0045, which means that simply reordering the layers and training over the whole corpus impacts little. Press et al. (2019) also reported this for machine translation, but they stopped here.

This seems to be a negative answer. However, we take a further step and ask one more question: Does different data favor different ordered layers? That is, we investigate whether each specific data has its own preference for one particular order. Intuitively, putting various data patterns in one order should not be the best choice. For example, harder samples may favor a particular order while easier ones favor another one. Thus, for each order, we count the ratio of samples that achieve the best score with that order. In Table 1, we find they almost lie on a uniform distribution (e.g., 17.9% samples achieve the best BLEU with order $SA{\rightarrow}ED{\rightarrow}FF$). Besides, we calculate the BLEU variance for each sample, and average all these variances, the result is 114.76, which is much larger than above corpus variance (0.0045). These both mean the data indeed has its own preference to different orders. In Table 2, we present translations from all decoders for on example with BLEU and TER score to give an evidence.

Motivated by above observations, in this work, we present Instance-wise Ordered Transformer (IOT), in which the layer order is determined by the specific data through instance-wise learning. To achieve this, we utilize a light predictor to predict the confidence for each order, given the corresponding classification losses as training signals. However, directly training the predictor with conventional (i.e., NMT) loss tends to quickly converge to a bad order, and ignore explorations on others. Thus, we introduce an exploration loss and an exploitation loss to make an effective training while keeping an unambiguous prediction for each data so that the best order can be decided during inference.

We evaluate our approach on 3 sequence generation tasks, including neural machine translation (NMT), abstractive summarization (ABS) and code generation (CG). For NMT, we work on 8 IWSLT and 2 WMT tasks, both on low-resource and rich-resource scenarios. Our method can consistently obtain 1.0 BELU score improvements over Transformer. For ABS, IOT also outperforms Transformer and other baselines on Gigaword dataset. For CG tasks, the results on 2 large-scale real-world code datasets (Java and Python) collected from Github surpass the state-of-the-art performances. These all demonstrate the effectiveness of our IOT. Furthermore, we provide detailed studies to verify that the instance-wise learning and order selection make a reasonable and necessary modeling.

The contributions of this work can be summarized as follows:

- We are the first to leverage instance-wise learning for layer order selection in a Transformer model (with shared parameters), and we demonstrate the instance-wise learning is critical.

- We demonstrate our learning approach can be universally applied to other structures beside Transformer (e.g., Dynamic Convolutions), as long as there are multiple different layers.

- Experiments on 3 sequence generation tasks and 9 datasets verify the effectiveness of IOT with consistent performance improvements.

## 2 RELATED WORK

**Architecture Exploration** Inventing novel architectures by human designing or automatic searching plays an important role in deep learning. Specific to Transformer structures, various modifications have been proposed. For example, human knowledge powered designs include DynamicConv (Wu et al., 2019), Macaron Network (Lu et al., 2019), Reformer (Kitaev et al., 2020) and others (Fonollosa et al., 2019; Ahmed et al., 2017; Shaw et al., 2018). As for automatic searching, neural architecture

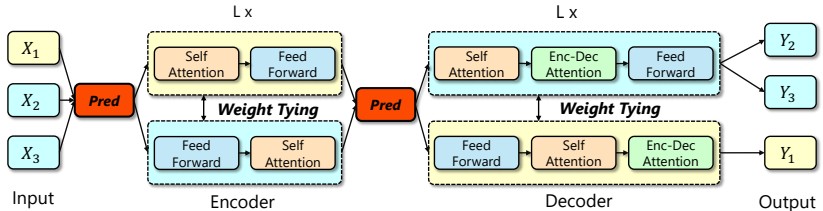

Figure 1: The IOT framework. *Pred* means the light predictor introduced in 3.1 for order selection. We show two ordered encoders/decoders here. After taking $X_1, X_2, X_3$, the selected order for $Y_2, Y_3$ is the lower encoder and upper decoder, while for $Y_1$ is the upper encoder and lower decoder.

search can discover networks with state-of-the-art performances but always with complicated computation, i.e., Evolved Transformer (So et al., 2019). The underlying principle is to add or replace some components of Transformer. For instance, Wu et al. (2019) replace self-attention with dynamic convolution, So et al. (2019) add a separate convolution layer in a new branch. Different from them, we, instead, only focus on the selection of layer orders for each data sample so as to improve the model performance, without a heavy modification. Besides, our approach is structure agnostic, which can be universally applied to other structures, only if multiple different layers exist.

**Instance-wise Learning** Deep learning models are trained over large-scale datasets, and data samples are often treated equally without modeling the difference between them. Some works attempt to weight each data with different importance (Ren et al., 2018; Hu et al., 2019; Chang et al., 2017) or feed data with curriculum learning according to its difficulty (Bengio et al., 2009; Fan et al., 2018). However, they often explicitly manipulate the data during training only, while no distinction exists in inference, and under one fixed model. Elbayad et al. (2020) take a step further and propose the depth-adaptive Transformer, which can forecast different depths of the network by predicting the required computation for a particular data. Similarly, Liu et al. (2020) propose a sample-wise adaptive mechanism to dynamically calculate the number of required layers. They both aim at reducing the computation cost and speed up the inference. Schwartz et al. (2020), Bapna et al. (2020) and Shazeer et al. (2017) all leverage conditional computation for each sample to control the computation and accuracy tradeoff during inference. Instead, we pay attention to the variant modeling functions and perform instance-wise order selection in order to boost the Transformer performance.

The most related work is Press et al. (2019), which manually generates randomly ordered Transformer encoders and finds the *Sandwich Transformer* can slightly reduce the perplexity of language modeling. However, they find that Sandwich Transformer pattern has no effect on NMT task. Besides, it still performs over the whole corpus without considering each specific data. We, instead, investigate on various sequence-to-sequence generation tasks and greatly improve the task performances through instance-wise learning, so as to discover the optimal ordered Transformer for each particular data.

## 3 INSTANCE-WISE ORDERED TRANSFORMER

The overall framework of IOT is presented in Figure 1. In comparison with the standard Transformer, IOT only incorporates light-weighted predictors and reorders the encoder/decoder with weight tying, under the constraint of almost same number of parameters and exempt from heavy modifications. In this section, we introduce the details of IOT, including training, inference and discussions.

**Notations** Sequence-to-sequence learning aims to map one sequence $x = [x_1, x_2, ..., x_{T_x}]$ into another sequence $y = [y_1, y_2, ..., y_{T_y}]$, where $x_i, y_j$ denotes the $i$-th and $j$-th token of $x$ and $y$, $T_x$ and $T_y$ are the corresponding lengths. Given one sentence pair $(x, y)$ and a learning model $\mathcal{M}$, we can define the training objective as minimizing the cross-entropy loss $\mathcal{L}_{\mathcal{M}} = -\sum_{j=1}^{T_y} \log P(y_j|y_{<j}, x)$. Besides, $D_{KL}(\mathbb{P}\|\mathbb{Q})$ denotes the Kullback-Leibler (KL) divergence between distributions $\mathbb{P}$ and $\mathbb{Q}$.

### 3.1 INSTANCE-WISE ENCODER/DECODER

IOT intends to break the fixed order of layers in Transformer. As shown in introduction, simply reordering the layers w.r.t the whole corpus impacts little, while each data has its own preference to orders. Therefore, IOT incorporates instance-wise learning to adjust the favorable order for each data.

As shown in Figure 1, both encoder and decoder in IOT consist of several blocks of SA, ED, FF layer with dynamic order, and we assume there are $M$ (e.g., $M = 2$) ordered encoders and $N$ (e.g., $N = 6$) ordered decoders (with shared weights). Inspired by the fact that lower training loss implies the higher proficiency confidence for candidate orders, we utilize the cross-entropy loss as signals to learn the confidence. That is, we calculate confidence $\gamma_m$ and $\lambda_n$ for each encoder $\mathtt{enc}_m$, decoder $\mathtt{dec}_n$ (resulted model $\mathcal{M}^{m,n}$), and use them to weight the training loss $\mathcal{L}_\mathcal{M}{}^{m,n}$. To calculate the confidence, we add a simple and light predictor to help distinguish the orders.

**Training** Given one source sequence $x = [x_1, x_2, ..., x_{T_x}]$, we first map each token into word embedding $e = [e_1, e_2, ..., e_{T_x}]$, where $e_i \in \mathbb{R}^d$, and then apply one light encoder predictor $\pi_{\mathtt{enc}}$ to predict the confidence of encoder orders using sentence embedding $s_e = \frac{1}{T_x} \sum_{i=1}^{T_x} e_i$. Concretely, $\pi_{\mathtt{enc}}$ takes $s_e$ as input and predicts $\gamma_m$ for $\mathtt{enc}_m$ by Gumbel-softmax (Jang et al., 2016):

$$\gamma_m = \frac{\exp\left((\log\left(\pi_{\mathtt{enc}_m}\right) + g_m\right)/\tau_e\right)}{\sum_{k=1}^M \exp\left((\log\left(\pi_{\mathtt{enc}_k}\right) + g_k\right)/\tau_e\right)}, \quad \pi_{\mathtt{enc}} = \mathrm{softmax}\left(s_e W_e\right), \tag{1}$$

where $g_m$ is sampled from Gumbel distribution: $g_m = -\log(-\log U_m), U_m \sim \mathrm{Uniform}(0, 1)$, $W_e \in \mathbb{R}^{d \times M}$ is the weight matrix, $\tau_e$ is a constant temperature to control the distribution to be identical approximation with categorical distribution. Simultaneously, the token embeddings $e$ will feed to the encoders to get hidden states $h = [h_1, h_2, ..., h_{T_x}]$, then we can calculate decoder order confidence $\lambda_n$ by one predictor $\pi_{\mathtt{dec}}$ in the same way as $\pi_{\mathtt{enc}}$:

$$\lambda_n = \frac{\exp\left((\log\left(\pi_{\mathtt{dec}_n}\right) + g_n\right)/\tau_d\right)}{\sum_{k=1}^N \exp\left((\log\left(\pi_{\mathtt{dec}_k}\right) + g_k\right)/\tau_d\right)}, \quad \pi_{\mathtt{dec}} = \mathrm{softmax}\left(s_d W_d\right), \tag{2}$$

where $s_d = \frac{1}{T_x} \sum_{i=1}^{T_x} h_i$ and $W_d$ is the weight matrix. For each ordered path through $\mathtt{enc}_m$ and $\mathtt{dec}_n$, we can obtain the training loss $\mathcal{L}_\mathcal{M}{}^{m,n}$, and the final cross-entropy loss is weighted by confidence $\gamma_m$ and $\lambda_n$ with $\mathcal{L}_\mathcal{M}{}^{m,n}$, formulately as:

$$\mathcal{L}_\mathcal{C} = \sum_{m=1}^M \sum_{n=1}^N (\gamma_m \cdot \lambda_n) \mathcal{L}_\mathcal{M}{}^{m,n}. \tag{3}$$

**Inference** During inference, we directly replace the Gumbel-softmax used in training with $\mathrm{argmax}$, in order to choose the most capable encoder and decoder for each sequence $x$:

$$\mathtt{enc} = \mathrm{argmax}\left(s_e W_e\right), \quad \mathtt{dec} = \mathrm{argmax}\left(s_d W_d\right). \tag{4}$$

**Discussion** The decoding process is almost the same as standard Transformer, with only little overhead for order predictions. One may concern the training cost is increased through our training. As we present in Section 5.1, the cost is actually affordable with a fast convergence. Currently, we reorder the layers of the encoder/decoder block and stack the same ordered block $L$ times (see Figure 1). A complex extension is to reorder all $L$ blocks of encoder/decoder and we take it as future work.

## 3.2 AUXILIARY LOSSES

As we can see, the predictors are trained in an unsupervised way, and we observe they lean to be lazy so that all samples quickly converge to one same order during training, without a senseful learning. Thus, to make an effective training and inference, we introduce exploration and exploitation losses.

(1) **Exploration**: first, we explore the diverse capability of all orders with help of a loss $\mathcal{L}_\mathcal{D}$ to encourage all orders to participate in training. The spirit is the same as to encourage exploration in reinforcement learning. The expected $\mathrm{softmax}$ probability $\mathbb{E}_x[\pi_x]$ (encoder/decoder) from the predictor should approximate the uniform distribution $\mathbb{Q} = [\frac{1}{N}, \frac{1}{N}, \ldots, \frac{1}{N}]$ (e.g., decoder orders), and we achieve this by minimizing KL-divergence between the statistical average $\mathbb{E}_x[\pi_x]$ and $\mathbb{Q}$:

$$\mathcal{L}_\mathcal{D} = D_{KL}(\mathbb{Q}\|\mathbb{E}_x[\pi_x]) = -\frac{1}{N} \sum_{n=1}^N \log(\mathbb{E}_x[(\pi_x)_n]) - \log N, \tag{5}$$

where $(\pi_x)_n$ is the probability of $n$-th decoder order for data $x$. For encoder order, it is $(\pi_x)_m$ processed in a same way as decoder.

(2) **Exploitation**: different from $\mathcal{L}_\mathcal{D}$ to keep all orders effectively trained, during inference, the output distribution $\pi_x$ for each data should be able to make an unambiguous $\mathrm{argmax}$ selection. We then introduce another loss $\mathcal{L}_\mathcal{S}$ to constrain each $\pi_x$ to be far away from the uniform distribution $\mathbb{Q}$. Concretely, we maximize the KL-divergence between each probability $\pi_x$ and $\mathbb{Q}$:

$$\mathcal{L}_\mathcal{S} = -\mathbb{E}_x\left[D_{KL}(\mathbb{Q}\|\pi_x)\right] = -\mathbb{E}_x\left[-\frac{1}{N}\sum_{n=1}^{N}\log(\pi_x)_n - \log N\right]. \tag{6}$$

Note that we clamp the value of probability $\pi_x$ since the KL value is theoretically unbounded. With above auxiliary losses, the final training objective is to minimize:

$$\mathcal{L} = \mathcal{L}_\mathcal{C} + c_1\mathcal{L}_\mathcal{D} + c_2\mathcal{L}_\mathcal{S}, \tag{7}$$

where $c_1$ and $c_2$ are coefficients to make a trade-off between $\mathcal{L}_\mathcal{D}$ and $\mathcal{L}_\mathcal{S}$. In this way, we can achieve effective training, while keeping the ability to distinguish the favorable order for each data.

**Discussion** $\mathcal{L}_\mathcal{D}$ and $\mathcal{L}_\mathcal{S}$ aim to keep effective training and unambiguous inference. There are several alternatives. The first is to simply decay the temperature $\tau$ in Equation (1) and (2), and remove the auxiliary losses. However, we do not notice obvious gain. Second is to linearly decay $c_1$ only and remove $\mathcal{L}_\mathcal{S}$, which is able to fully train all orders at the beginning and loose this constraint gradually. We find this is also beneficial, but our two losses method performs better.

## 4 EXPERIMENTS

We conduct experiments on 3 sequence generation tasks: neural machine translation (both low-resource and rich-resource), code generation and abstractive summarization. The main settings of each experiment are introduced here, and more details can be found in Appendix A.

### 4.1 DATASET

**Neural Machine Translation** For the low-resource scenario, we conduct experiments on IWSLT14 English↔German (En↔De), English↔Spanish (En↔Es), IWSLT17 English↔French (En↔Fr), English↔Chinese (En↔Zh) translations. The training data includes $160k$, $183k$, $236k$, $235k$ sentence pairs for each language pair respectively. For the rich-resource scenario, we work on WMT14 En→De and WMT16 Romanian→English (Ro→En) translations. For WMT14 En→De, we filter out $4.5M$ sentence pairs for training and concatenate newstest2012 and newstest2013 as dev set, newstest2014 as test set. For WMT16 Ro→En, we concatenate the $0.6M$ bilingual pairs and $2.0M$ back translated data[2] for training, newsdev2016/newstest2016 serve as dev/test set.

**Code Generation** Code generation aims to map natural language sentences to programming language code. We work on one Java (Hu et al., 2018) and one Python dataset (Wan et al., 2018), following Wei et al. (2019) to process the two datasets. The Java dataset is collected from Java projects on Github, and the Python dataset is collected by Barone & Sennrich (2017). We split each dataset with ratio $0.8 : 0.1 : 0.1$ as training, dev and test set.

**Abstractive Summarization** Abstractive summarization is to summarize one long sentence into a short one. The dataset we utilized is a widely acknowledged one: Gigaword summarization, which is constructed from a subset of Gigaword corpus (Graff et al., 2003) and first used by Rush et al. (2017). The training data consists of $3.8M$ article-headline pairs, while the dev and test set consist of $190k$ and $2k$ pairs respectively.

### 4.2 MODEL AND OPTIMIZATION

For IWSLT translation tasks, we use `transformer_iwslt_de_en` setting as model configuration. The number of block, embedding size and feed-forward network (FFN) size are 6, 512, 1024. WMT tasks use `transformer_vaswani_wmt_en_de_big` configuration, with 6 blocks, embedding size 1024 and FFN size 4096. Optimization and learning scheduler are the default settings in Vaswani et al. (2017). For code generation, block number/embedding size/FFN size are 3, 256, 1024

---

[2]http://data.statmt.org/rsennrich/wmt16_backtranslations/ro-en/.

Table 3: Preliminary results of varied orders on IWSLT14 De→En task.

(a) Results of encoder, decoder orders, and their combinations.

|  | De→En |
| --- | --- |
| Transformer | 34.64 |
| Encoder ($M = 2, N = 1$) | 35.18 |
| Decoder ($M = 1, N = 6$) | 35.60 |
| Encoder×Decoder ($M = 2, N = 6$) | 35.24 |
| Encoder×Decoder ($M = 2, N = 4$) | 35.30 |
| Encoder×Decoder ($M = 2, N = 2$) | 35.25 |

(b) Results of varied number of decoder orders.

|  | De→En |
| --- | --- |
| Transformer | 34.64 |
| IOT ($M = 1, N = 6$) | 35.60 |
| IOT ($M = 1, N = 5$) | 35.65 |
| IOT ($M = 1, N = 4$) | 35.62 |
| IOT ($M = 1, N = 3$) | 35.58 |
| IOT ($M = 1, N = 2$) | 35.32 |

Table 4: BLEU scores of IOT on eight IWSLT low-resource translation tasks.

|  | En→De | De→En | En→Fr | Fr→En | En→Zh | Zh→En | En→Es | Es→En |
| --- | --- | --- | --- | --- | --- | --- | --- | --- |
| Transformer | 28.57 | 34.64 | 35.9 | 36.1 | 26.3 | 18.4 | 39.0 | 40.6 |
| IOT | 29.52 | 35.62 | 37.2 | 37.8 | 27.2 | 19.3 | 40.1 | 41.7 |

respectively. Others are the same as NMT. For summarization, we take `transformer_wmt_en_de`, with 6 blocks, embedding size 512 and FFN size 2048. Dropout (Srivastava et al., 2014) is set to be 0.3. Other settings are also the same as NMT task. Implementation is developed on Fairseq (Ott et al., 2019). We first grid search $c_1$, $c_2$ on IWSLT14 De→En dev set, and then apply them on other tasks. The best setting is $c_1 = 0.1$, $c_2 = 0.01$, and the importance study of $c_1$, $c_2$ is shown in Appendix B.1.

## 4.3 EVALUATION

We use `multi-bleu.perl` to evaluate IWSLT14 En↔De and all WMT tasks for a fair comparison with previous works. For other NMT tasks, we use `sacre-bleu` for evaluation. During inference, we follow Vaswani et al. (2017) to use beam size 4 and length penalty 0.6 for WMT14 En→De, beam size 5 and penalty 1.0 for other tasks. For code generation, the evaluation is based on two metrics, the sentence BLEU computes the n-gram precision of a candidate sequence to the reference, and the percentage of valid code (PoV) that can be parsed into an abstract syntax tree (AST). As for summarization, the generated summarization is evaluated by ROUGE-1/2/L F1 score (Lin, 2004).

## 4.4 MAIN RESULTS

**Encoder/Decoder Orders** Encoder block only contains SA and FF layers, the resulted max number of encoder layer orders $M$ is 2, while for decoder, the max order variants $N$ is 6. Therefore, we first evaluate the utilization of encoder orders, decoders orders, and both orders on IWSLT14 De→En translation, in order to see the impacts of different number of order candidates and their combinations. In Table 3 (a), we can see that 2 ordered encoders improve the result, and 6 ordered decoders achieve more gain. This meets our expectation, since the search space is limited when there are only 2 ordered encoders. However, if we train both encoder and decoder orders (e.g., $M = 2, N = 6$), the results (e.g., 35.30) can not surpass the 6 decoders only (35.60). We suspect the search space is too large so that training becomes hard, and decoder orders play a more important role than encoder orders for sequence generation. Therefore, we turn to investigate different decoder order candidates (refer to Appendix A.3 for detailed combinations) in Table 3 (b). Results show that $N = 4, 5, 6$ achieve similar strong performances (results on other tasks/datasets are in Appendix A.4). Thus, considering the efficiency and improvements, we utilize $N = 4$ ordered decoders (order $1, 2, 4, 6$ in Table 1) to reduce training cost in later experiments.

**NMT Results** BLEU scores on 8 IWSLT low-resource tasks are shown in Table 4. As we can see, IOT achieves more than 1.0 BLEU points improvement on all tasks (e.g., 1.7 on Fr→En). The consistent gains on various language pairs well demonstrate the generalization and effectiveness of our method. We then present comparison with other works on IWSLT14 De→En task in Table 5 (a), and IOT is also better than several human designed networks. The results of WMT14 En→De and WMT16 Ro→En are reported in Table 6. We also compare with existing works, such as the unsupervised Ro→En based on pre-trained cross-lingual language model (Lample & Conneau, 2019).

Table 5: Results on IWSLT14 De→En translation task (a), Java and Python code generation tasks (b).

(a) Results on IWSLT14 De→En translation.

| Method | BLEU |
|---|---|
| Transformer | 34.64 |
| **IOT** | **35.62** |
| Adversarial MLE (Wang et al., 2019a) | 35.18 |
| DynamicConv (Wu et al., 2019) | 35.20 |
| Macaron Network (Lu et al., 2019) | 35.40 |
| MADL (Wang et al., 2019c) | 35.56 |

(b) Results on Java and Python code generations.

| Method | Java | | Python | |
|---|---|---|---|---|
| | BLEU | PoV | BLEU | PoV |
| Transformer | 24.58 | 74.44% | 13.20 | 61.89% |
| **IOT** | **25.51** | **77.30%** | **14.05** | **63.14%** |
| Wei et al. (2019) | 17.17 | 27.4% | 12.09 | 51.9% |

Similarly, our method outperforms them and shows our framework can work well on rich-resource scenario.

**Code Generation Results** The results are shown in Table 5(b). We can observe that Transformer obtains better result than the LSTM-based work (Wei et al., 2019). Compared with Transformer, IOT can further improve the quality of generated code. Specifically, IOT boosts Transformer with $0.93$ BLEU/$2.86\%$ PoV gain on Java generation and $0.75$ BLEU/$1.25\%$ PoV gain on Python respectively. Again, these results well demonstrate the effectiveness of our method.

**Abstractive Summarization Results** The IOT performances on summarization task are shown in Table 7. From the results, we can see IOT achieves $0.8$, $0.7$ and $1.0$ scores gain of ROUGE-1, ROUGE-2 and ROUGE-L metrics over standard Transformer on Gigaword summarization. IOT also surpasses other works such as reinforcement learning based method (Wang et al., 2018), which again verifies our approach is simple yet effective.

| Method | En→De |
|---|---|
| Transformer* | 29.12 |
| **IOT** | **30.03** |
| Shaw et al. (2018) | 29.20 |
| Ott et al. (2018) | 29.30 |
| Wu et al. (2019) | 29.70 |
| So et al. (2019) | 29.80 |

| Method | Ro→En |
|---|---|
| Transformer* | 37.73 |
| **IOT** | **38.83** |
| Sennrich et al. (2016) | 33.90 |
| Lample & Conneau (2019) | 38.50 |

Table 6: WMT14 En→De and WMT16 Ro→En translation results. *stands for our reproduced result.

| Method | ROUGE-1 | **ROUGE-2** | ROUGE-L |
|---|---|---|---|
| Transformer (Vaswani et al., 2017) | 35.59 | 17.74 | 32.98 |
| **IOT** | 36.37 | **18.46** | 33.89 |
| RNNSearch+MRT (Ayana et al., 2016) | 36.54 | 16.59 | 33.44 |
| Concept pointer+DS (Wang et al., 2019b) | 37.01 | 17.10 | 34.87 |
| RNNSearch+select+MTL+ERAML (Li et al., 2018) | 35.33 | 17.27 | 33.19 |
| CGU (Lin et al., 2018) | 36.30 | 18.00 | 33.80 |
| Reinforced-Topic-ConvS2S (Wang et al., 2018) | 36.92 | 18.29 | 34.58 |

Table 7: ROUGE-1/2/L F1 scores for Gigaword summarization.

## 5 STUDY AND ANALYSIS

### 5.1 INFERENCE/TRAINING COST

As discussed before, our approach only increases negligible parameters and inference time cost. Here we compare the detailed inference time and model size of our framework to the standard Transformer. The detailed parameter numbers and inference time on IWSLT14 En↔De test set are shown in Table 8. Since we only add one linear layer and softmax layer as the predictor, the number of extra parameters is $M \times hidden\_size$ (encoder predictor) or $N \times hidden\_size$ (decoder predictor), which is negligible compared to other model parameters. Therefore, IOT introduces more model diversity and improves the performance, but under the constraint of almost same number of parameters. As for the inference time, the only difference is from the one-pass order prediction and the cost is extremely low compared with heavy autoregressive generation process, which can be seen from Table 8.

Apart from the inference cost, one may concern about the training cost since IOT trains multiple orders in one model. To see the influence, we provide several statistics here. Specifically, on the four IWSLT En→X translation tasks, we analyze the cost by counting the training time for each epoch, the epoch number when model convergences, and the corresponding total training time. The numbers

| Method | En→ De | | De→En | |
|---|---|---|---|---|
| | Inference time (s) | #parameters | Inference time (s) | #parameters |
| Transformer | 1487.19 | 36741120 | 1432.03 | 36741120 |
| IOT ($N = 2$) | 1496.70 | 36742144 | 1429.26 | 36742144 |
| IOT ($N = 3$) | 1470.40 | 36742656 | 1480.83 | 36742656 |
| IOT ($N = 4$) | 1505.83 | 36743168 | 1444.74 | 36743168 |
| IOT ($N = 5$) | 1491.40 | 36743680 | 1424.10 | 36743680 |
| IOT ($N = 6$) | 1479.20 | 36744192 | 1464.00 | 36744192 |

Table 8: Inference time and model parameters counted for Transformer and our framework on IWSLT14 En↔De. The study is performed on a single Tesla P100 GPU card.

| | | Transformer | IOT ($N = 2$) | IOT ($N = 3$) |
|---|---|---|---|---|
| **En→De** | **Epoch Time (s)** | 277.1 | 475.6(1.72×) | 685.4(2.47×) |
| | **Epoch Number** | 67 | 42(0.63×) | 39(0.58×) |
| | **Total Time (s)** | 18565.7 | 19975.2(1.08×) | 26730.6(1.44×) |
| **En→Fr** | **Epoch Time (s)** | 410.5 | 715.1(1.73×) | 1024.1(2.49×) |
| | **Epoch Number** | 55 | 37(0.67×) | 33(0.60×) |
| | **Total Time (s)** | 22577.5 | 26458.7(1.17×) | 33795.3(1.49×) |
| **En→Zh** | **Epoch Time (s)** | 349.9 | 612.3(1.75×) | 931.4(2.66×) |
| | **Epoch Number** | 59 | 34(0.58×) | 30(0.51×) |
| | **Total Time (s)** | 20644.1 | 20818.2(1.01×) | 27942.0(1.35×) |
| **En→Es** | **Epoch Time (s)** | 310.1 | 538.0(1.73×) | 784.4(2.53×) |
| | **Epoch Number** | 69 | 43(0.62×) | 30(0.43×) |
| | **Total Time (s)** | 21369.9 | 23134.0(1.08×) | 23532.0(1.10×) |

Table 9: Training cost analysis for Transformer and our IOT on four IWSLT translation tasks. The study is performed on a single Tesla P100 GPU card.

are presented in Table 9, and we can have several observations. Take IWSLT14 En→De translation as an example, (1) jointly optimizing different orders indeed introduces more training cost for each epoch. Transformer baseline costs $277.1s$ per epoch training, while our IOT costs $475.6s$ and $685.4s$ with $N = 2$ and $N = 3$ orders respectively, the increased cost ratio is about $1.72\times$ and $2.47\times$ (but less than 2.0 and 3.0). (2) However, we find that with the shared parameters between these orders, the model convergence also becomes faster. Transformer needs 67 epochs when converge, while our IOT only needs $42(0.63\times)$ and $39(0.58\times)$ epochs for $N = 2$ and $N = 3$ orders, much fewer than Transformer. (3) The total training cost actually is not increased much. IOT ($N = 2$) and IOT ($N = 3$) are about $1.08\times$ and $1.44\times$ training time compared with Transformer baseline (the ratio for IOT ($N = 3$) is only 1.10 on IWSLT17 En→Es). From these observations, we can see that the increased training cost is affordable due to the fast convergence.

## 5.2 CASE VERIFICATION

We perform a study with $N = 3$ to verify that IOT has made a necessary instance-wise order selection. We first split IWSLT14 En↔De dev set into 3 subsets according to the prediction of $\pi_{\text{dec}}$, and then we decode each subset use all 3 ordered decoders, and report the BLEU results. As shown in Figure 2, each subset indeed achieves the best score on the corresponding predicted order (outperforms other orders by 0.2-0.4 BLEU). We also do the same study on the test set, and the predicted order outperforms others by 0.7-0.8 BLEU. These well prove that IOT makes a reasonable prediction.

Besides, we find that the predicted orders correlate to different sentence difficulties. In our case, the set 1 sentences belong to decoder 1 achieve highest BLEU than other sets, which means set 1 is relatively simple to translate, and vice versa for samples in set 2. These imply that different difficulty sentences have different structure preferences. We provide statistics and examples in Appendix B.2.

## 5.3 APPLY ON ANOTHER STRUCTURE (DYNAMICCONV)

As we discussed, our instance-wise layer reordering is structure agnostic. In this subsection, we evaluate this by applying our approach on DynamicConv network (Wu et al., 2019) beyond standard

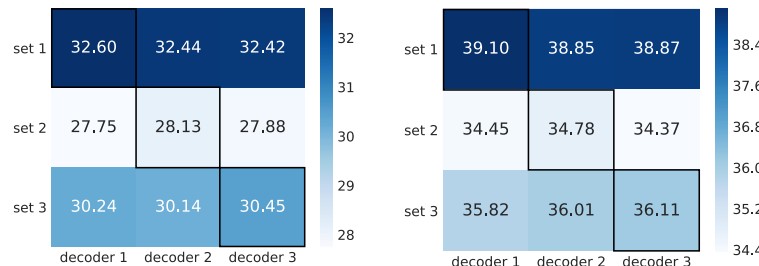

(a) BLEU scores on IWSLT14 En→De. (b) BLEU scores on IWSLT14 De→En.

Figure 2: BLEU scores of three subsets (divided by predicted decoder) on all decoders.

Transformer, which replaces the self-attention with dynamic convolution. We train layer ordered DynamicConv on $N = 2, 3, 4$ decoders and test the performances. The BLEU score of standard DynamicConv is $35.20$, and with our instance-wise order learning, we achieve $35.60, 35.82, 35.87$ for $N = 2, 3, 4$ ordered decoders respectively (near $0.7$ point gain). Therefore, this study verifies our claim that our approach can be applied to other structures, as long as multiple different layers exist.

## 5.4 DISCUSSIONS

**Ensemble** Since our framework involves multiple orders (with shared parameters), which is also done in ensemble framework, we make a comparison with ensemble. The ensemble method trains multiple models with different parameters separately in an independent way. While our work trains orders in a joint way with an intention to make them more diverse. More importantly, from the view of time and memory cost, the ensemble framework increases $N$ times which is totally different from ours. In this sense, our method can be combined with ensemble to further boost performance. The competitive results on IWSTL14 En↔De test set are shown in Table 10. We can clearly conclude that IOT and ensemble are complementary to each other.

| Ensemble Models | En→De | De→En |
|---|---|---|
| 1-model (standard) | 28.57 | 34.67 |
| 2-model (standard) | 29.71 | 35.92 |
| 3-model (standard) | 30.08 | 36.40 |
| 4-model (standard) | 30.18 | 36.54 |
| 1-model (IOT) | 29.52 | 35.62 |
| 2-model (IOT) | 30.38 | 36.80 |
| 3-model (IOT) | 30.93 | 37.25 |
| 4-model (IOT) | 31.02 | 37.38 |

Table 10: Ensemble performances of standard Transformer and IOT.

**Regularization** IOT consists of different ordered blocks in a weight tying method, which may looks like a parameter regularization to some extent. However, we show that IOT is more than regularization and can be complementary with other regularization methods. **Setting (1):** We first train a Transformer model on IWSLT14 De→En task with all shared decoder orders, but without instance-wise learning, and test the performance with each order. We find the BLEU scores on test set are near $34.80$ for each order, much worse than IOT, which means that simply regularizing the shared parameters for different orders is not the main contribution to performance improvement, and our instance-wise learning is critical. **Setting (2):** Another experiment is that we train Transformer with LayerDrop (Fan et al., 2019), a dropout technique to regularize the layer parameters. The test BLEU is $35.40$, which achieves about $0.8$ score improvement over Transformer. After applying IOT with LayerDrop, we obtain further gains than IOT only ($35.62$) to reach a BLEU score $36.13$. Therefore, this demonstrates IOT is not only regularization and can be smoothly integrated with other regularization methods. More details and experiments on other tasks are shown in Appendix B.3.

## 6 CONCLUSION

In this work, we propose Instance-wise Ordered Transformer, which leverages instance-wise learning to reorder the layers in Transformer for each data. Compared with standard Transformer, IOT only introduces slightly increased time cost. Experiments on 3 sequence generation tasks and 9 datasets demonstrate the effectiveness of IOT. We also verify that our approach can be universally applied to other structures, such as DynamicConv. In future, we plan to work on more complicated reordering in each block, as well as other tasks such as multi-lingual translation and text classification.

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

## A    Experimental Settings and More Results

### A.1    Detailed Data Settings

**Neural Machine Translation** Following the common practice (Ott et al., 2019), we lowercase all words for IWSLT14 En↔De. For IWSLT14 En↔De, En↔Es, IWSLT17 En↔Fr, we use a joint source and target vocabulary with $10k$ byte-pair-encoding (BPE) (Sennrich et al., 2015) operations, and for IWSLT17 En↔Zh, we use a seperate source and target vocabulary. For all WMT tasks, sentences are encoded by a joint source and target vocabulary of $32k$ tokens.

**Code Generation** In the Java dataset, the numbers of training, validation and test sequences are $69, 708$, $8, 714$ and $8, 714$ respectively, and the corresponding numbers for Python are $55, 538$, $18, 505$ and $18, 502$. All samples are tokenized. We use the downloaded Java dataset without further processing, and use Python standard AST module to further process the python code. The source and target vocabulary sizes in natural language to Java code generation are $27k$ and $50k$, and those for natural language to Python code generation are 18k and $50k$. In this case, following Wei et al. (2019), we do not apply subword tokenization like BPE to the sequences.

**Abstractive Summarization** The Gigaword corpus represents for a headline generation task, each source article contains about $31.4$ tokens on average, while the target headline contains near $8.3$ tokens per sentence. The training data consists of $3.8M$ article-headline pairs, while the validation and test set consist of $190k$ and $2k$ pairs respectively. We preprocess the dataset in a same way as NMT task. The words in the source article and target headline are concatenated to make a joint BPE vocabulary. After preprocessing, there are $29k$ subword tokens in the vocabulary.

### A.2    Detailed Model/Training Configurations

**Model Configuration** The detailed model configurations are as follows:

- `transformer_iwslt_de_en` setting: 6 blocks in encoder and decoder, embedding size 512, feed-forward size 1024, attention heads 4, dropout value 0.3, weight decay 0.0001.

- `transformer_vaswani_wmt_en_de_big` setting: 6 blocks in encoder and decoder, embedding size 1024, feed-forward size 4096, attention heads 16, dropout value 0.3, attention dropout 0.1, relu dropout 0.1.

- `transformer_wmt_en_de_big` setting: 6 blocks in encoder and decoder, embedding size 0124, feed-forward size 4096, attention heads 16, dropout value 0.3.

**Optimization** We adopt the default optimization setting in Vaswani et al. (2017). Adam (Kingma & Ba, 2014) optimizer with $\beta_1 = 0.9, \beta_2 = 0.98$ and $\epsilon = 10^{-9}$. The learning rate scheduler is `inverse_sqrt` with warmup steps $4, 000$, default learning rate is 0.0005. Label smoothing (Szegedy et al., 2016) is used with value 0.1. As introduced, to learn the predictors, we clamp the `softmax` output with value 0.05.

### A.3    Results of Order Combinations

We show in the paper that different number of orders (e.g., $N = 4$ or $N = 5$) have varied performances. Therefore, one necessary point is about the different combinations of these $N$ decoders. Here, we work on $N = 5$ IOT model to show the results of different order candidates.

We first present each ordered decoder in Table 11 again (same as in Table 1). For the $N = 5$ ordered decoders with IOT model, we show the performances with 5 combined orders selected from all six variants on dev set of IWSLT14 De→En and En→De translations. The results are reported in Table 12. We can see the different combinations achieve similar strong performances, which shows that our

| **Code** | 1 | 2 | 3 | 4 | 5 | 6 |
|---|---|---|---|---|---|---|
| **Order** | SA→ED→FF | FF→SA→ED | ED→FF→SA | ED→SA→FF | SA→FF→ED | FF→ED→SA |

Table 11: Each numbered code for one specific ordered decoder.

| IOT ($N = 5$) | 12345 | 12346 | 12356 | 12456 | 13456 | 23456 |
|---|---|---|---|---|---|---|
| **En→De** | 30.64 | 30.65 | 30.53 | 30.58 | 30.70 | 30.67 |
| **De→En** | 36.71 | 36.67 | 36.72 | 36.79 | 36.74 | 36.75 |

Table 12: BLEU scores for IWSTL14 De→En and En→De translations on dev set. '12345' represents the combinations of order $1, 2, 3, 4, 5$ decoders.

approach is robust towards different order combinations. This also demonstrate that the importance of IOT is the diversity among order candidates that can help each data distinguish them. For other $N$ ordered decoders, the patterns are similar. Therefore, here we only report the $N$ combinations used for IOT experiments in the paper as follows: IOT ($N = 2$) is combined by order $4, 6$ (*ED→SA→FF* and *FF→ED→SA*), and IOT ($N = 3$) is order $1, 4, 6$, IOT ($N = 4$) is order $1, 2, 4, 6$, and IOT ($N = 5$) is order $1, 2, 4, 5, 6$.

### A.4 RESULTS OF DIFFERENT NUMBER OF DECODERS

The results of $N = 4$ ordered decoders (order $1, 2, 4, 6$) are mainly reported in the paper. Here, we also show results of other $N$ decoders for all tasks, along with the Transformer baseline.

The results of different $N$ decoders for WMT14 En→De and WMT16 Ro→En translations, code generation task, and Gigaword summarization are reported in Table 14, Table 15 and Table 16 respectively. As we can see, more ordered decoders can bring better performance, which supports the effectiveness of our framework and demonstrates the data has its own favor towards different orders. Considering the efficiency, we do not perform experiments with more than 4 decoders for these tasks.

| Model | En→De | De→En | En→Fr | Fr→En | En→Zh | Zh→En | En→Es | Es→En |
|---|---|---|---|---|---|---|---|---|
| Transformer | 28.57 | 34.64 | 35.9 | 36.1 | 26.3 | 18.4 | 39.0 | 40.6 |
| IOT ($N = 6$) | 29.48 | 35.60 | 37.4 | 37.7 | 27.1 | 19.2 | 40.2 | 41.5 |
| IOT ($N = 5$) | 29.51 | 35.65 | 37.2 | 37.6 | 27.2 | 19.2 | 40.2 | 41.9 |
| IOT ($N = 4$) | 29.52 | 35.62 | 37.2 | 37.8 | 27.2 | 19.3 | 40.1 | 41.7 |
| IOT ($N = 3$) | 29.43 | 35.58 | 37.0 | 37.6 | 27.0 | 19.1 | 39.7 | 41.5 |
| IOT ($N = 2$) | 29.18 | 35.32 | 36.6 | 37.1 | 26.8 | 18.9 | 39.6 | 41.0 |

Table 13: BLEU scores on 8 IWSLT tasks with different $N$ ordered decoders (with shared weights).

| Model | WMT14 En→De | WMT16 Ro→En |
|---|---|---|
| Transformer | 29.12 | 37.73 |
| IOT ($N = 2$) | 29.71 | 38.57 |
| IOT ($N = 3$) | 29.89 | 38.79 |
| IOT ($N = 4$) | 30.03 | 38.83 |

Table 14: BLEU scores for WMT14 En→De and WMT16 Ro→En translation tasks of different $N$ ordered decoders in IOT.

| | Java | | Python | |
|---|---|---|---|---|
| Model | BLEU | PoV | BLEU | PoV |
| Transformer | 24.58 | 74.44% | 13.20 | 61.89% |
| IOT ($N = 2$) | 25.44 | 77.88% | 13.97 | 62.22% |
| IOT ($N = 3$) | 25.51 | 75.83% | 14.00 | 62.04% |
| IOT ($N = 4$) | 25.51 | 77.30% | 14.05 | 63.14% |

Table 15: BLEU and PoV scores for Java and Python code generation results of different $N$ ordered decoders in IOT.

| Model | ROUGE-1 | **ROUGE-2** | ROUGE-L |
|---|---|---|---|
| Transformer | 35.59 | 17.87 | 32.98 |
| IOT ($N = 2$) | 36.14 | 18.37 | 33.61 |
| IOT ($N = 3$) | 36.15 | 18.48 | 33.81 |
| IOT ($N = 4$) | 36.37 | 18.46 | 33.89 |

Table 16: ROUGE F1 scores for Gigaword abstractive summarization results of different $N$ ordered decoders in IOT.

| | $c_1/c_2$ | 0.0 | 0.005 | 0.01 | 0.05 |
|---|---|---|---|---|---|
| **IOT ($N = 2$)** | 0.0 | 35.97 | 36.36 | 36.46 | 36.42 |
| | 0.05 | 36.35 | – | 36.58 | – |
| | 0.1 | 36.31 | 36.56 | **36.60** | 36.54 |
| | 0.5 | 36.39 | – | 36.40 | – |
| **IOT ($N = 3$)** | 0.0 | 36.16 | 36.44 | 36.66 | 36.51 |
| | 0.05 | 36.47 | – | 36.47 | – |
| | 0.1 | 36.35 | 36.48 | **36.79** | 36.57 |
| | 0.5 | 36.44 | – | 36.62 | – |

Table 17: BLEU scores for IWSLT14 De→En dev set. The performances are varied by different weighted auxiliary losses controlled by $c_1$ and $c_2$ value.

# B MORE STUDIES

## B.1 IMPACT OF WEIGHTED AUXILIARY LOSSES

We conduct another study on IWSLT14 De→En dev set to investigate the impact of our proposed auxiliary losses controlled by weight $c_1$ and $c_2$. The values of $c_1$ and $c_2$ are varied between $[0.0, 0.05, 0.1, 0.5]$ and $[0.0, 0.005, 0.01, 0.05]$ respectively, and the results are presented in Table 17. It can be seen that the best configuration is $c_1 = 0.1$ and $c_2 = 0.01$. Therefore, we report the leading results in the paper with $c_1 = 0.1, c_2 = 0.01$. The results also clearly demonstrate that the two additional losses are necessary to make our framework effective.

## B.2 DATA EXAMPLES VERIFICATION

As discussed in Section 5.2, the data split by the corresponding predicted order is in different pattern. For example, the difficulty of each set is different. We therefore analyze the split data and calculate some statistics among these subsets. Specifically, we first count the sentence number $S$, the tokens $T$, and the distinct vocabulary $D_i$ in each subset. We show these numbers in Table 18, along with corresponding averaged BLEU score (see Figure 2). We can see that the vocabulary size of set 1 is the smallest, and set 2 is the largest, which means there are more distinct words in set 2. This leads the generation of set 2 to be harder than set 1, which maps the BLEU score ranking among these sets. Besides, we also calculate the token frequency $f_{ij}$ for token $j$ in each own subset $i$, and sum the frequency of top 20 tokens in each subset, $F_i = \sum_{j=1}^{20} f_{ij}$, to give another evidence. The results also show that $F_1$ is the highest, which means the tokens in set 1 contains most frequent words to make an easy learning, while set 2 is harder since $F_2$ is small.

| Set | $S_i$ | $T_i$ | $D_i$ | $F_i$ | $BLEU_{Avg}$ |
|---|---|---|---|---|---|
| $i = 1$ | 2,404 | 56,036 | 4,557 | 0.3693 | 32.49 |
| $i = 2$ | 2,093 | 49,077 | 4,897 | 0.3234 | 27.92 |
| $i = 3$ | 2,786 | 66,226 | 4,699 | 0.3668 | 30.28 |

Table 18: Statistics of each English valid subset on IWSLT14 En→De translation. $S_i$ is the number of sentences in set $i$. Correspondingly, $T_i$ is the token number, $D_i$ is the vocabulary size. $F_i = \sum_{j=1}^{20} f_{ij}$ is the sum of the frequency of top 20 tokens in set $i$, where $f_{ij}$ is the frequency for token $j$.

| Set 1 | i will come to it later .
i wasn 't very good at reading things . |
|-------|-----------------------------------------------------------------------|
| Set 2 | this is a very little known fact about the two countries .
it 's why they leave lights on around the house . |
| Set 3 | and how can we do that ?
it 's our faith , and we will be lo@@ y@@ al to it . |

Table 19: Samples of each English valid subset on IWSLT14 En→De translation.

We further take a look at the data and find the sentences in set 1 are mostly "simple sentences", and set 2 contains many "emphatic sentences", while set 3 is somehow mixed. In Table 19, we provide some sentence examples belong to each subset to give more clarifications.

## B.3 REGULARIZATION

In Section 5.4, we have provided an example of regularization experiments on IWLST14 De→En translation, which demonstrates that our IOT is not only regularization and can be smoothly integrated with other regularization methods. To give more evidences and more details, we extend the regularization experiments on all IWSLT translation tasks (IWSLT14 En↔De, IWSLT14 En↔Es, IWSLT17 En↔Zh, IWSLT17 En↔Fr), and WMT16 Ro→En translation. The two specific settings of the experiments are as follows. **Setting (1):** The first experiment is "ordered Transformer" without instance awareness. That is, all the reordered architectures are trained on the whole same corpus with equal weights, and the parameters for these reordered architectures are shared. More specifically, the decoder block has different ways to order *SA*, *ED*, and *FF* layers (e.g., *FF→SA→ED*, *SA→ED→FF*, etc), but the parameters for the reordered blocks are shared. Mathematically, the loss function is: $\mathcal{L}_\mathcal{C} = \sum_{n=1}^{N}(\lambda_n \cdot \mathcal{L}_\mathcal{M}^n)$, where $\mathcal{L}_\mathcal{M}^n$ is the model loss function for $n$-th ordered decoder. Compared with Eqn (3), the weight $\lambda_n$ is fixed to be 1 here. At inference, we first find out the best order according to the dev performance and apply it on the test set. We cannot use instance-wise reordered model in this setting, while our proposed IOT can. The experiments are conducted with `transformer_iwslt_de_en` configuration for IWSLT translations, and `transformer_vaswani_wmt_en_de_big` configuration for WMT16 Ro→En translation. **Setting (2):** We integrate another regularization technique 'LayerDrop' (Fan et al., 2019) into both the Transformer baseline and our IOT ($N = 4$) method, while other settings remain unchanged. The study results of these two settings are represented in Table 20.

From the results, we have same conclusions as discussed in Section 5.4. Simply sharing the parameters of different decoders as a regularization cannot boost the model performance ("Transformer + (1)" in Table 20), while our IOT can further improve the performance with other regularization methods.

## B.4 ROBUSTNESS

An impact of IOT training besides performance gain is that the model can be more robust compared to one order only. In Table 21, we provide one example to prove the robustness. We train one Transformer model by decoder order *1*, and to decode the sentences with all orders in inference. Obviously, only decoding with order *1* leads to good performance, while other orders can not achieve reasonable scores since the layer order is changed and the feature exaction becomes incorrect. As for IOT, the generated sequences remain stable and high results for each order.

## B.5 VISUALIZATION

To better understand the difference between IOT and standard Transformer, we investigate on the training process and provide visualization results about model optimization and performance improvements. Specifically, we plot the curve of training loss, validation loss, as well as the validation BLEU score and test BLEU score along the training epochs on IWSLT14 De→En translation dataset. The loss curves are visualized in Figure 3, and the BLEU curves are presented in Figure 4.

From the validation loss curves of Figure 3(b) and 3(c), we can first see that our IOT ($N = 3$) training converges faster than Transformer baseline and shows the advantage of IOT, which is consistent to

| Model | En→De | De→En | En→Fr | Fr→En | |
|---|---|---|---|---|---|
| Transformer | 28.57 | 34.64 | 35.9 | 36.1 | |
| IOT ($N = 4$) | 29.52 | 35.62 | 37.2 | 37.8 | |
| Transformer + (1) | $28.35 \pm 0.12$ | $34.68 \pm 0.05$ | $35.9 \pm 0.23$ | $36.7 \pm 0.14$ | |
| Transformer + (2) | 29.01 | 35.40 | 36.2 | 36.8 | |
| IOT ($N = 4$) + (2) | 29.94 | 36.13 | 37.4 | 38.1 | |
| **Model** | **En→Zh** | **Zh→En** | **En→Es** | **Es→En** | **Ro→En** |
| Transformer | 26.3 | 18.4 | 39.0 | 40.6 | 37.73 |
| IOT ($N = 4$) | 27.2 | 19.3 | 40.1 | 41.7 | 38.83 |
| Transformer + (1) | $26.4 \pm 0.05$ | $18.8 \pm 0.17$ | $37.7 \pm 0.28$ | $39.6 \pm 0.11$ | $37.82 \pm 0.05$ |
| Transformer + (2) | 26.8 | 19.0 | 39.4 | 40.8 | 38.33 |
| IOT ($N = 4$) + (2) | 27.3 | 19.5 | 40.6 | 42.5 | 38.98 |

Table 20: Regularization study experiments on 8 IWSLT translation tasks and WMT16 Ro→En translation. We study both setting (1): train Transformer model with all shared decoders but without instance-wise learning, and setting (2): add LayerDrop (Fan et al., 2019) regularization technique experiments on these tasks.

| Order | Transformer | IOT |
|---|---|---|
| $1^\star$ | 35.84 | 36.42 |
| 2 | 27.96 | 36.45 |
| 3 | 8.05 | 36.47 |
| 4 | 5.11 | 36.38 |
| 5 | 1.32 | 36.46 |
| 6 | 0.35 | 36.41 |

Table 21: Robustness study on IWSTL14 De→En translation task on dev set. $^\star$ is the order trained by Transformer.

our analysis in Section 5.1. The converged (smallest) validation loss value seems to be similar to Transformer baseline, but please note that the loss computation of IOT is different from Transformer baseline. As shown in Eqn (3), the loss function of IOT is a weighted sum of loss values for each order, while for Transformer, it is only one order loss. Therefore, when we turn to the comparison of validation BLEU score, the superiority of our IOT can be clearly verified. From the BLEU score curves in Figure 4, it is obvious that IOT achieves better BLEU score than standard Transformer along the whole training epochs, on both validation and test sets. These visualized results well demonstrate the effectiveness of our IOT approach.

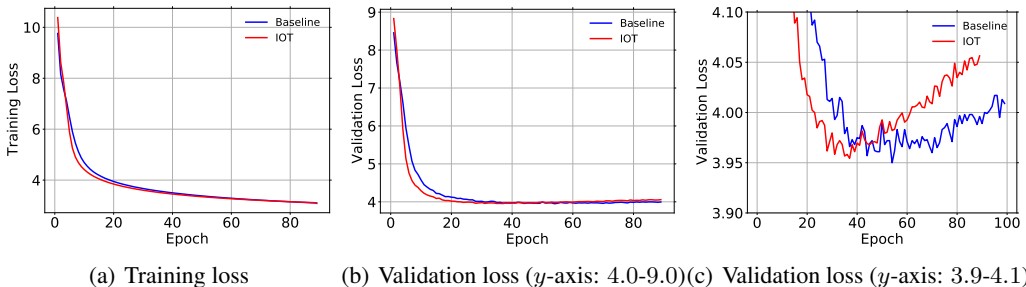

(a) Training loss     (b) Validation loss ($y$-axis: 4.0-9.0)(c) Validation loss ($y$-axis: 3.9-4.1)

Figure 3: Comparison of training/validation loss curves along the model training on IWSLT De→En translation. 'IOT' is IOT ($N = 3$) and 'Baseline' is Transformer. Figure 3(c) is the same curve as 3(b), except the value of $y$-axis in Figure 3(c) is between 3.9-4.1, while 4.9-9.0 for Figure 3(b).

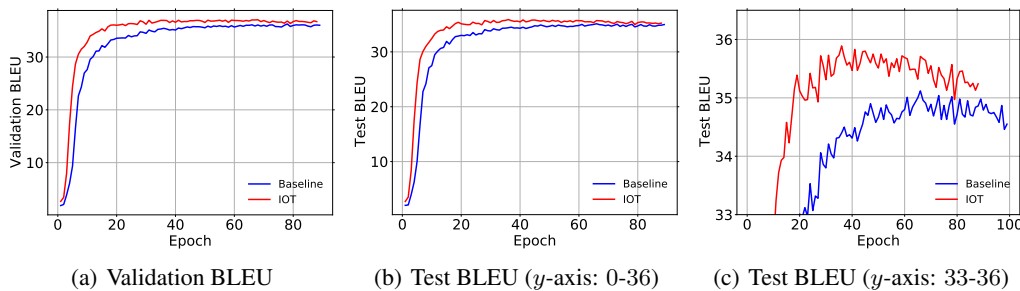

(a) Validation BLEU            (b) Test BLEU ($y$-axis: 0-36)            (c) Test BLEU ($y$-axis: 33-36)

Figure 4: Comparison of validation/test BLEU curves along the model training on IWSLT De→En translation. 'IOT' is IOT ($N = 3$) and 'Baseline' is Transformer. Figure 4(c) is the same curve as 4(b), except the value of $y$-axis in Figure 4(c) is between 33-36, while 0-36 for Figure 4(b).

