# OpenReview forum: "IOT: Instance-wise Layer Reordering for Transformer Structures"
_ICLR.cc/2021/Conference — ICLR 2021 Poster_

### Official Review · AnonReviewer4 · 2020-10-21
**overclaim**

**Rating:** 5
**Confidence:** 5

**Review:**

This paper studies the influence of the arrangement order for the internal structure in a single-layer Transformer (they named it as layer order) on the performance. It makes a hypothesis that different layer order has an impact on the performance of the model, and the hypothesis is verified by experiments. Based on this hypothesis, a lightweight layer order predictor is designed to predict an input-related layer order, and through reinforcement learning with two auxiliary loss, the model can not only be trained by diverse layer order, but also make unambiguous layer order prediction as far as possible. The IOT structure proposed in this paper has been evaluated on several datasets of machine translation, abstract summarization and code generation. Compared with the traditional transformer structure, it has been improved consistently, which shows the effectiveness of the proposed structure.



My concerns are as the following,

1. Training time: it seems that this paper only reports the time of inference. However, compared with the standard Transformer, the IOT proposed in this paper only changes the order inside the Transformer layer and it does not increase the additional inference time consumption. In addition, too simple predictor will have little impact over time cost. However, for training, due to the additional loss, additional order exploration, etc., the convergence time of the model may be changed. Therefore, I expect to see comparison of IOT with Transformer.


2. Internal order: since the mentioned internal structure order of Transformer will affect the results, meanwhile, according to the previous work, the order of LayerNorm may also affect the performance of the model. I am curious if such a structure can be also considered and compared.

3. Model capacity: There is a claim that the capacity of the model has been improved due to the introduction of IOT, while there is no direct evidence from experiments that can support such a claim. Since the parameters of the model have not changed, I feel it hard to regard the improved model capacity. Besides, it is not clear on the definition of the model capacity.

4. Different input and different structures: because this paper claims that different inputs should bring different structures, but according to the current training method with mini-batch for neural networks, it seems that it is impossible to give a completely different structure for each sentence. I expect to see some obvious enough evidence on different structures in sequence-level or batch-level from such training.

Minor comments,

It would be better if the ablation experiments are done on a larger dataset such as WMT 14.

---

> ### Author Response · Authors · 2020-11-19
> **Response to Reviewer4**
>
> Thanks for your review and comments.  For your concerns, we make responses as follows.
> 1.	As for the training time, we have reported some results in Appendix B.1 previously. In the revised version, we add more analysis in Section 5.1 of the main text, please kindly check it again. We have compared the number of epochs for convergence, training time per epoch, and the total training time. The quick conclusion is that, though each training epoch takes more time, our algorithm requires fewer epochs to converge. Therefore, the overall cost does not increase much. For example, IOT (N=3) only takes $1.44\times$ total training time on IWSLT14 En->De translation compared with Transformer baseline.
> 2.	About LayerNorm order. Previous works [1, 2, 3] have demonstrated the difference between pre-norm and post-norm of LayerNorm. Pre-norm is more suitable to train a very deep model, while for the same number of layers (e.g., 6 layers, which are not very deep) in Transformer, post-norm empirically achieves better accuracy. Since our goal is to improve model accuracy without adding extra parameters, we adopt the post-norm in our IOT.
> 3.	About model capacity. Indeed, it is hard to strictly and mathematically define “model capacity”. We have revised this, as shown in abstract and related parts. What we want to express is that our method has more flexibility to build functions for sequence generation. Take translation as an example. For the standard Transformer, there is only one fixed order of layers (e.g., SA->ED->FF) for the translation function (i.e., translation model). In comparison, our IOT can build variant models with different layer orders, and the order is adaptively determined based on the input instance. Therefore, each sentence has more opportunities to select the optimal one among these functions so as to improve the performance.
> 4.	About the different structures for each sentence during training. Actually, during training, the layer order of each sentence is represented by a distribution (output by $\pi$) instead of a deterministic one. Each sentence in a mini-batch has its own distribution of the orders. Therefore, the difference of the structure selection is reflected by the different predicted distribution for each sentence during training. Indeed, we are not “to give a completely different structure for each sentence” during training (though we never claim this in the paper). It is automatically decided by the predicted distribution, and these structures are weighted (weight is the predicted distribution) and trained together. To give empirical evidence, we show the predicted distribution values along the training epochs of two training sentence examples from a same mini-batch (on IWSLT14 De->En training set) as follows. We can see that the distributions become more and more concentrated as training goes on, which indicts the different structures selected by different sentences.
> |  Sentence A:    Epoch    |     Order 1    |     Order 4    |     Order 6    |
> |---------------------------|----------------|----------------|----------------|
> |     1       |    0.31  |   0.38    |     0.31    |
> |     15     |    0.26   |   0.29    |     0.45    |
> |     25     |    0.09   |   0.14    |     0.76    |
> |     35     |    0.02   |   0.02    |     **0.96**    |
> |---------------------------|----------------|----------------|----------------|
> |  **Sentence B**:   **Epoch**    |     **Order 1**    |     **Order 4**    |    **Order 6**    |
> |---------------------------|----------------|----------------|----------------|
> |     1      |    0.42    |    0.32    |   0.26   |
> |     15    |    0.54    |    0.24    |   0.22   |
> |     25    |    0.83    |    0.10    |   0.07   |
> |     35     |     **0.97**   |   0.02   |   0.01 |
> |---------------------------|----------------|----------------|----------------|
> 5.	About the ablation study on WMT 14 dataset. First, this is a big dataset, and each training consumes much time. It will cost a lot for ablation studies. Therefore, we conduct ablation studies only on IWSLT14 small dataset. We believe similar results can be observed on WMT14 as well. Second, performing ablation studies on small datasets such as IWSLT14 is widely adopted and accepted in many works, for example, [5, 6, 7]. Third, for the study on numbers of different orders, we have provided results in Table 14 in Appendix A.4. We will add other ablation studies in later version.
>
> Hope the above responses address your concerns.
>
> [1] On Layer Normalization in the Transformer Architecture, ICML, 2020.\
> [2] Learning Deep Transformer Models for Machine Translation, ACL, 2019.\
> [3] Understanding the Difficulty of Training Transformers, EMNLP, 2020. \
> [4] The Tradeoffs of Large Scale Learning, NeurIPS, 2007.\
> [5] Non-autoregressive Neural Machine Translation, ICLR, 2018.\
> [6] Incorporating BERT into Neural Machine Translation, ICLR, 2020.\
> [7] Incorporating BERT into Parallel Sequence Decoding with Adapters, NeurIPS, 2020.

---

> > ### Comment · AnonReviewer4 · 2020-11-20
> > **More training time is kept as concern**
> >
> > Thanks for your feedback. Most of them are satisfactory.
> > I read all the other reviews as well. Now I would like to clarify my last concern, which is still unsolved.
> > I am curious about the true performance improvement all the time. As well known that both more parameters (i.e., larger model) and longer training (i.e., more epochs, or each epochs taken longer) can always bring better performance, which is not too surprising. There are many technique improvement in deep learning, in which most of them actually exploited such two tricks in an obscure way.
> > For this work, my doubt on the training time is shown true,  i.e., 1.44 times of the original one (thanks for reporting this), which is not insignificant.
> > I did not say all work which actually apply longer training and larger model are valueless, however, I would not appericiate such work too much and keep alert on their presentation. Therefore, to better understand this work, I may suggest the authors can further provide a training curve comparison between the the proposed IOT and its baseline showing the loss (or BLEU) vs. training epochs.

---

> > > ### Author Response · Authors · 2020-11-21
> > > **Response to follow up concern about training loss/BLEU curve**
> > >
> > > Thanks for reading all the reviews and acknowledging our responses.
> > >
> > > For the last concern about the performance improvement and training time. Though 1.44x cost is not insignificant for IOT (N=3) on IWSLT14 En->De, in table 9, we also provide that IOT (N=2) only costs 1.08x and 0.6 BLEU improvement is obtained (shown in Table 13). By the way, for IWSLT17 En->Es, IOT (N=3) only costs 1.10x with 1.0 BLEU improvement. According to your great suggestion, we plot the loss curves of training/valid loss comparison of IOT and Transformer in Appendix B.5 (we have updated the revised paper), and the curves of validation/test BLEU scores along the training epochs on IWSLT14 De->En translation (both Transformer and IOT are trained for 80 epochs, longer than the converged epochs as a response to your point about 'more epochs'). From Figure 3 and 4 in Appendix B.5, we can see that our IOT performs better and achieves higher BLEU score than Transformer along the whole training epochs, on both validation and test sets, which means IOT achieves true performance improvement all the time. Also, from the Figures, we can demonstrate that the improvement/advantage of IOT is not from longer training, 'more epochs or each epoch taken longer'. Kindly have a check of Appendix B.5.

---

### Official Review · AnonReviewer2 · 2020-10-27
**A simple modification of Transformer that consistently improve its performance without parameters/computation drawback.**

**Rating:** 7
**Confidence:** 4

**Review:**

This paper aims at improving the transformer architecture by reordering the encoder/decoder layers. Improving the performance of transformer is an active area of research and has numerous applications. The authors first show that a single layers' ordering is not better than all others and further demonstrate that per instance layer ordering with parameter sharing consistently improve overall performance (which is quite surprising). Other works considered per instance early return (depth) but I am not aware of per instance layer reordering.

#### Strong points
+ The layer order decision is made on simple sentence embeddings which does not add significant computation compared to the transformer.
+ The weights are shared between layers of different reordering which does not increase the number of parameters.
+ The authors conduct extensive evaluation on Neural Machine Translation, Code Generation and Summarization and show consistent improvement of their method (IOT).
+ They show that this method is not a form of ensembling/experts (which was my main concern), IOT is orthogonal to ensembling (Section 5.3).
+ The same reordering trick can be applied to other multi-layer architectures.

#### Weak points

- The different layer orders process the samples differently and require to split a batch into smaller batches for each ordering.
- Why some samples are better processed by a specific layer ordering remains not understood.

#### Decision

I tend to accept this paper as the method is novel and the results are good. The method can be applied without strong drawbacks in term of computation or number of parameters.

#### Questions

- Table 15: I am surprised that inference time is not affected by the batching per order (see weak point), did you apply some tricks like masking to be more GPU friendly?
- Can you show the proportion of each ordering? (for instance IOT (N=6) on IWSLT14) I am curious if it is balanced and if the decisions made by the classifier are consistent with the best performing order in Table 1 when each transformer order is trained separately.

---

> ### Author Response · Authors · 2020-11-19
> **Response to Reviewer2**
>
> Thanks for your acknowledgment and review comments. We provide the following responses for the comments and questions.
> *	Yes, to make batch inference and save the inference cost, we have to implement a masking matrix (supported by `torch.index_select()`) for the batch data and let each ordered decoder to only decode the corresponding data samples. Besides, the masked data are distributed on different CUDA streams (supported by `torch.cuda.Stream()` ) to run concurrently. The time reported in Table 8 is the sum of per sample decoding on the test set.
> *	Thanks for your great question. As shown in Appendix B.3, the different subsets of sentences indeed have different patterns to favor different layer orders. For example, order $1$ is favored by relative “simple sentences” while order $4$ is favored by “emphatic sentences”. Though we have those shallow observations, unfortunately, we have no deep understanding on the internal relationship between a specific layer order and a certain data pattern. It is hard to analyze such a deep neural network, we will continue working on this.
> *	Yes, this is also an interesting point to show the consistence. We count the proportion as you suggested on an IOT (N=6) model compared with Table 1, and the corresponding ratio for each decoder is as follows: 0.18518, 0.12758, 0.19841, 0.19415, 0.14817, 0.14651. Overall speaking, we can see it is balanced as in Table 1. As for the consistent decisions between Table 1 and IOT, assume that we independently train $N$ models with different layer orders. For each test data $x$, we translate $x$ using the above models, and calculate the corresponding sentence-level BLEU scores. Let $O(x,k)$ denote a $k$-element set of layer orders, whose corresponding models can achieve the top-$k$ highest BLEU score of sentence $x$ ($k\le N$). Let $A(x)$ denote the layer order determined by our IOT for sequence $x$. We calculate the top@K accuracy (consistence) as follows:$$\text{top@k}= \frac{\sum_{x\in\mathcal{X}}I(A(x)\in O(x,k))}{\vert \mathcal{X}\vert},$$ where $\mathcal{X}$ denotes the collection of all test inputs. Note that the top@1 accuracy measures the “the best performing order” as you mentioned. We conduct experiments on IWSLT’14 De->En. The top-k accuracies are listed in the below table. The results show that in most cases, IOT provide a relatively good order (at least, top-3 with 84.1%) of the Transformer model. Note that as reported in Table 1 of the main text, the variance of sentence BLEU is large. Therefore, it is hard to find the exact top-1 order but we can still achieve 31.3% accuracy. We further calculate the corpus BLEU that each sentence is decoded by the order with corresponding top/second/third BLEU. The top-1 order BLEU score is 42.48, and the second, third order BLEU is 38.48 and 35.84 (close to our IOT BLEU 35.62), much higher than Transformer baseline 34.7. From these results, we can see that our IOT indeed makes reasonable and good decisions to choose the best/better orders as in Table 1.
> |Top-$k$|top@k |
> |--------|-------|
> | Top-1 | 31.3% |
> | Top-2 | 61.6% |
> | Top-3 | 84.1% |
>
>
> We hope the above discussions address your questions.

---

> > ### Comment · AnonReviewer2 · 2020-11-24
> > **Thank you for your answer**
> >
> > I would like to thank the authors for their detailed answer. They addressed my concerns.

---

### Official Review · AnonReviewer3 · 2020-10-28
**Good paper, regularization experiments could be more thorough**

**Rating:** 7
**Confidence:** 4

**Review:**

Summary: This work explores instance-wise layer re-ordering in transformers. The key idea is to incorporate classifiers that predict the ordering of sub-layers (self-attention, cross-attention, feed-forward) from the averaged input sequence representation, one classifier each for the encoder and the decoder. During training the model uses a soft Gumbel-noised output of the classifier to combine the outputs from stacks with differently ordered sub-layers. During inference the argmax of the classifier prediction is used to generate the output sequence. The model is trained with two auxiliary losses: (i) A loss to ensure the expected output of the classifiers is uniform and (ii) A loss to ensure the classifier output for each individual sample is distant from uniform.
Experiments are performed on 8 IWSLT translation tasks, code generation and Gigaword summarization. The proposed approach results in significant gains over a standard transformer on all the reported tasks.

Strengths:
1. Significant gains on a variety of sequence-to-sequence NLP tasks.
2. The authors conduct several experiments trying to uncover the source of the quality gains, comparing against ensembling approaches, layer re-ordering based regularization and layerdrop on the IWSLT En-De task. The approach is also demonstrated to improve the quality of a DynamicConv based Machine Translation model (although less effective than in the Transformer setting).

Weaknesses:
1. While the authors do compare against regularization (layer re-ordering), the settings of these experiments are not very clear and the results are reported only on the IWSLT De->En task. A more careful comparison on all the reported tasks to counter against the regularization hypothesis would significantly strengthen the claims in the paper.

Other questions for authors:
1. On page 2 in the introduction, the reported average Bleu variance per sample (114.76) seems way too high, given that Bleu $\in [0, 100]$.
2. Do the predictor outputs approach the argmax? If not, is there a quality loss incurred by choosing the argmax instead of using the expectation like during training?
3. Did the authors try using a single predictor for both the encoder and decoder (i.e. a single classifier predicting $M\times N$ classes)?
4. Are the layer orders preferred by a particular input (or the subset of input examples that prefer the same layer order) consistent across randomly initialized training runs? This analysis might help justify the claim that certain layer orders are better for harder/easier inputs.
5. Other relevant work on instance-level adaptation in NLP: [1,2,3]

Recommendation: I think there are several interesting ideas in the paper, and would recommend (weak) acceptance. There is still some doubt as to whether the quality gains are arising from a regularization effect or the instance-based layer re-ordering. I would be willing to update my recommendation given additional evidence to address this concern.

References:

[1] OUTRAGEOUSLY LARGE NEURAL NETWORKS: THE SPARSELY-GATED MIXTURE-OF-EXPERTS LAYER, Shazeer et al.

[2] Controlling Computation versus Quality for Neural Sequence Models, Bapna et al.

[3] The Right Tool for the Job: Matching Model and Instance Complexities, Schwartz et al.

EDIT: Updated recommendation to acceptance (7) following author response.

---

> ### Author Response · Authors · 2020-11-19
> **Response to Reviewer3 (Part 1)**
>
> Thanks for your comments and acknowledgment, below we provide our responses to your concerns and questions.
>
> Towards the weaknesses:
> 1. Sorry about the unclear part of the experiment settings. We have updated the main text and Appendix in revised version. We describe clearer settings of this paragraph here. (1): The first experiment is “ordered Transformer” without instance awareness. That is, all the reordered architectures are trained on the whole same corpus with equal weights, and the parameters for these reordered architectures are shared. More specifically, the decoder block has different ways to order SA, ED, and FF layers (e.g., SA->ED->FF, FF->SA->ED, etc), but the parameters for the reordered blocks are shared. Mathematically, the loss function is: $\mathcal{L_{C}} = \sum_{n=1}^{N}(\lambda_n \cdot \mathcal{L_{M}}^{n})$, where $\mathcal{L_{M}}^{n}$ is the model loss function for $n$-th ordered decoder. Compared with Eqn (3), the weight $\lambda_n$ is fixed to be $\frac{1}{N}$ here. At inference, we first find out the best order according to the dev performance and apply it on the test set. We cannot use instance-wise reordered model for translation in this setting, while our proposed IOT can. We conduct experiments on iwslt14 De->En translation with transformer_iwslt_de_en configuration. We find the BLEU score for each decoder is similar and near 34.80, which is far behind our IOT performance. This demonstrates that simply regularizing the shared parameters for different orders is not the main contribution to performance improvement, and it verifies our instance-wise learning is critical. (2) The second regularization experiment is to integrate another regularization technique “LayerDrop” [1] into both the Transformer baseline and our IOT (N=4) method, while other settings remain unchanged. By using LayerDrop, the test BLEU of standard Transformer can be boosted from 34.64 to 35.40, and that of IOT can be boosted from 35.62 to 36.13. This demonstrates IOT is orthogonal to the existing regularization techniques and can be smoothly integrated with them.
> 2.	As your suggestion to give a comparison on all reported tasks, we provide all IWSLT translation tasks (8 directions translations) and WMT16 Ro->En translation here. The results of setting (1) and (2) described in response 1 on these tasks are as follows. We can see the improvements are consistent on IWSLT14 De->En setting. We also provide these results in Appendix B.3 of the revised paper.
> |                        |     Ende              |     Deen            |     Enfr          |     Fren          |     Enzh           |     Zhen           |     Enes           |     Esen           |     RoEn           |
> |------------------------|-----------------------|---------------------|-------------------|-------------------|--------------------|--------------------|--------------------|--------------------|--------------------|
> |     Transformer        |     28.57             |     34.64           |     35.9          |     36.1          |     26.3           |     18.4           |     39.0           |     40.6           |     37.73          |
> |     IOT(N=4)           |     29.52             |     35.62           |     37.2          |     37.8          |     27.2           |     19.3           |     40.1           |     41.7           |     38.83          |
> |     Transformer+(1)    |     28.35$\pm$0.12    |     34.68 $\pm$0.05    |     35.9$\pm$0.23    |     36.7$\pm$0.14    |     26.4$\pm$0.05     |     18.8$\pm$0.17     |     37.7$\pm$0.28     |     39.6$\pm$0.11     |     37.82$\pm$0.05    |
> |     Transformer+(2)    |     29.01             |     35.40           |     36.2          |     36.8          |     26.8           |     19.0           |     39.4           |     40.8           |     38.33          |
> |     IOT(N=4)+(2)       |     29.94             |     36.13           |     37.4          |     38.1          |     27.3           |     19.5           |     40.6           |     42.5           |     38.98          |
>
>
> Towards the questions:
> 1.	What we report in the paper is the variance instead of standard derivation. Therefore, the value could exceed 100. We also report the standard derivation here. The average BLEU variance per sample is 114.76, which is calculated by: $var_{avg} = \frac{1}{|D|}\sum_{x\in D}var(BLEU(x))$, and the average BLEU standard derivation per sample is 7.28, which is calculate by: $ std\_{avg} = \frac{1}{|D|}\sum_{x\in D}std(BLEU(x))$. The score is large for each data sample because BLEU is sensitive to the n-gram differences. Take the example on Table 2, the variance for this example is 1063.18, and the standard derivation is 32.61, which is large.
>
> (continue to Part 2)

---

> > ### Author Response · Authors · 2020-11-19
> > **Response to Reviewer3 (Part 2)**
> >
> > 2.	Yes, the translation quality is impacted by the argmax predictor outputs. There is still a performance gap between predictor output distribution and hard argmax. We make an experiment to see the quality. Specifically, we leverage all ordered decoders to perform an “internal ensemble” decoding, and then test the BLEU score, the result is 35.89 BLEU score, which is slightly higher than our IOT argmax decoding (35.62), but the inference speed of IOT is much faster than ensemble decoding.
> > 3.	We have tried a single predictor for both encoder and decoder order selection, which uses the embedding vectors of each sentence to learn the predictor. The test BLEU on IWSLT14 De->En is 35.20, which is similar to the numbers in Table 3(a), and worse than current modeling (35.62). We suspect the embeddings are not as representative enough as the encoder outputs.
> > 4.	Thanks for this interesting question. We conduct a study to train two randomly initialized IOT (N=3) models and analyze the data samples with preferred orders, and count the decision overlap between the two runs. We find that the average overlap ratio is about 82.6%, which means most of the decisions are consistent.
> > 5.	Thanks for suggesting the related works, we have included these papers in revised version.
> >
> > We hope the above discussions address your questions.
> >
> > [1] Reducing transformer depth on demand with structured dropout, ICLR, 2019.

---

### Official Review · AnonReviewer1 · 2020-10-29
**Does this increase the training flops by a factor of n_reorder?**

**Rating:** 5
**Confidence:** 4

**Review:**

I think the general idea behind this paper is very exciting: architectures mutating in an instance-based way.

I was hoping to see that the authors had achieved this whilst adding only a small overhead of parameters and training FLOPs such that we could be sure any gains aren't due to augmenting these two quantities (i.e. instance-based re-ordering is the key ingredient) and to be sure this could be applied in the large-model setting and theoretically improve production translation systems.

However the proposed architecture is essentially training an ensemble of three, or n_reorder, weight-shared models and then at inference time, using a hard-threshold. The paper really tries to brush over this fact, stating there is negligible additional cost in the abstract and in several other parts of the text. There is only this one line to acknowledge the fact that we are training 3x (or even 4x) models:  "One may concern the training cost is increased through our training. As we present in Appendix B.1, the cost is actually acceptable with a fast convergence." and then this section in the appendix continues to mostly discuss inference cost, but finally admits the n_reorder flops increase stating that it is acceptable because the model converges faster. It doesn't seem likely the model will converge 3x faster in general, especially if it learns to mostly select only one re-ordering after some training.

My general sense is the proposed architecture sits in a very tenuous space. If the researcher has 3-4x space and flops to train a model 3-4x larger, then they should clearly do so. This will get much better performance. If the researcher has 3-4x space to train a larger model but wants faster inference, perhaps this approach could be useful but I would suspect training large and then pruning or distilling to be much more effective. If the architecture trained with hard re-ordering decisions and thus used the same number of flops, but still outperformed the baseline it would be a clear win.

Suggestions:
* Compare training flops and eval performance to the approach of training a 3x larger model that is then pruned or distilled.
* Be up-front about the training wall-clock time in the main text, instead of stating everything has negligible cost and only reporting inference speed.
* Consider using RL or another hard-decision approach.
* Consider comparing to a model that has the same number of parameters but 3x the compute (e.g. a transformer with 3x depth but shared weights for each of the three layers).

---

> ### Author Response · Authors · 2020-11-19
> **Response to Reviewer1 (Part 1)**
>
> Thanks for your reviews, comments, and suggestions.
>
> We understand your concerns about the training cost. Regarding our writing, we are not trying to brush the training cost fact but instead, we just want to emphasize more on the inference cost. First, the “negligible cost” mainly refers to the inference cost instead of training, and the additional parameter cost. We are sorry not to clearly state these in abstract and related parts, we have revised accordingly. Second, the training cost is indeed increased, as we stated in the paper, but the focus of this paper is to improve the model accuracy without adding additional inference cost. This is a realistic and crucial factor in production translation systems. For most production translation systems in the industry, to provide a quick translation inference service with high quality is much more important than caring about the training cost. From the research aspect, we also concern about the training cost, and therefore we conduct some statistical analysis. We only report in the appendix due to the space limitation of the main text previously, we have put into the main text in revised version. Yes, the convergence speed is not 3x, but the convergence indeed is much faster, and the additional total training cost is not much: for example, IOT (N=3) only costs $1.44\times$ total training time on IWSLT14 En->De translation compared with Transformer baseline model, as we reported in the revised version in Section 5.1 of main text. Hope these numbers and statistics address your concerns.
>
> For your concerns about architecture:
> 1.	As for the larger model and space, we have to say this is not always true to “get much better performance” with a 3-4x larger model (at least in seq2seq machine translation, MT). For example, [1,2,3] all report that simply training deeper/larger transformer models (for MT) is hard, either no obvious gain or training diverges. Successful training of deep/large Transformer models requires special or complex designs [2,4,5], the performances are not much better than ours (e.g., 30.1 in [5] v.s. our IOT 30.03 for WMT14 En->De translation), and the inference speed becomes slower for those methods.
> 2.	Knowledge distillation from a large teacher model can help improve the performance, but it does not bring as much improvement as our method. Besides, it consumes more training cost (See our response below for your suggestion 1).
> 3.	For the last scenario, RL based learning may achieve a “clear win” result to make a hard-reordered training with similar number of FLOPS, but it needs more trial-and-error, and the result may not be so good, please also refer to our response to your suggestion 3 below.
>
> We also conduct experiments according to your suggestions:
>
> * For Suggestion 1: We train 3x larger Transformer models for iwslt14 De->En translation through deeper and wider models, the performance for 3x deeper model is 33.85 (worse than Transformer baseline 34.64), while the model of 3x wider is failed to converge, which demonstrates the hardness of training such deep/wide models. As for further distillation, we adopt the sequence-level distillation method described in [6] to train a 1x student Transformer model (take 3x deeper model as teacher), the performance is 35.21, which surpasses the teacher model and the Transformer baseline, but it is still worse than IOT (35.62) and the inference costs much more than our IOT. We also count the training time of 3x deeper model and the distillation training of a 1x Transformer model, to further show the advantages of IOT. The training cost of 3x deep teacher model is 757.3s x 33 epochs = 24990.9s, the 1x student model training costs 516.9s x 30 epochs = 15507.0s, and thus the total cost is 24990.9s + 15507.0s = $40497.9s$. IOT (N=3) costs 699.4s x 35 epochs = $24479.0s$, which is only $0.6\times$ cost compared with training of 3x Transformer and distillation of 1x student. We summarize the statistics in the below table.
> | Model                     | BLEU  | Training Cost                 | Total Cost                     |
> |---------------------------|-------|-------------------------------|--------------------------------|
> | 3x deeper (teacher)       | 33.85 | 757.3s x 33 epochs = 24990.9s |                                |
> | 1x distillation (student) | 35.21 | 516.9s x 30 epochs = 15507.0s | 24990.9s + 15507.0s = 40497.9s |
> | IOT (N=3)                 | 35.62 | 699.4s x 35 epochs = 24479.0s | 24479.0s  ($0.6\times$)                     |
> * For Suggestion 2: Thanks for your suggestion! We have put the detailed number/statistics of the training and inference cost in Section 5.1 in the revised main text. It is much clearer now.
>
>
> (continue to Part 2)

---

> > ### Author Response · Authors · 2020-11-19
> > **Response to Reviewer1 (Part 2)**
> >
> > * For Suggestion 3: RL approach can perform training with same number of flops for each forward pass. However, the drawback is that final training cost will greatly increase when convergence, due to the trial-and-error, sampling operations, exploration and exploitation, and the result is hard to guarantee. We have an initial result on iwslt14 De->En translation through policy gradient RL training, the BLEU score is near 34.98 with order selection, which is slightly better than Transformer baseline (34.64) but worse than IOT (35.62). We will continue working on this.
> > * For Suggestion 4: We conduct experiments on iwslt14 De->En translation according to your description. We train a model with [SA, SA, SA, ED, ED, ED, FF, FF, FF] with shared weights in each block, and another model with [SA, ED, FF, SA, ED, FF, SA, ED, FF] setting. However, we find these settings are failed to converge, and the training losses are very high. These results show that the performance improvement of IOT is not from simply parameter sharing, deep models, or regularization.
> >
> > We hope the above answers help address your concerns.
> >
> > [1] Training Deeper Neural Machine Translation Models with Transparent Attention, EMNLP, 2018.\
> > [2] Depth Growing for Neural Machine Translation, ACL, 2019.\
> > [3] Understanding the Difficulty of Training Transformers, EMNLP, 2020.  \
> > [4] Learning Deep Transformer Models for Machine Translation, ACL, 2019.\
> > [5] Very Deep Transformers for Neural Machine Translation, arxiv, 2020.\
> > [6] Sequence-Level Knowledge Distillation, EMNLP, 2016.

---

### Author Response · Authors · 2020-11-19
**Paper Revision**

Dear Reviewers and AC,
We have updated our revised paper. The detailed revision includes:
1.	We provide more comparison of training/inference cost between IOT and Transformer baseline in Section 5.1.
2.	We provide more regularization experiments and clear descriptions in Appendix B.3, and we slightly update Section 5.4.
3.	We update the related work in Section 2.
4.	We revise the claim of model capacity in the paper.
5.	We add Appendix B.5 to visualize the loss curve and BLEU curve along with the training process.

---

### Decision · Program_Chairs · 2021-01-07
**Final Decision**

**Decision:**

Accept (Poster)

**Comment:**

The paper investigates the order of Transformer modules and its effect on performance. The proposed approach IOT, consists of several pre-defined encoder and decoders with different orderings (and weight sharing), along with a light predictor which is trained to choose the best configuration per instance.

Most reviewers found the general idea of predicting the order of Transformer modules at instance-level quite intriguing. Other strengths included wide range of evaluation tasks, major empirical gains, and novelty.
 R1 and R4 raise valid and important concerns on validity of results when the model size and training time are controlled.
However, after carefully reading the author response and the revised paper, I feel that this issue is resolved.
The authors provide comparison with larger models, ensemble models, and models trained longer, and in all cases the gains are still obvious.
 Overall, I feel that the general idea behind this paper is very exciting and could inspire more research in this direction. Therefore I recommend accept.